# OpenReview forum: "Steering at the Source: Style Modulation Heads for Robust Persona Control"
_ICML.cc/2026/Conference — ICML 2026 regular_

### Official Review · Reviewer_9JDa · 2026-03-09

**Soundness:** 2
**Presentation:** 3
**Significance:** 2
**Originality:** 3
**Overall Recommendation:** 4
**Confidence:** 4

**Summary:**

This paper investigates the coherency degradation problem in activation steering. Existing methods lead to a sharp decline in the coherence of generated text when enhancing target traits, and this degradation cannot be detected by traditional metrics such as MMLU and perplexity.
The authors hypothesize that the root cause of coherence degradation lies in the fact that the residual stream aggregates all information from previous layers, and intervention indiscriminately amplifies noise unrelated to the target features. Therefore, intervention should be carried out directly at the "source" of feature generation, rather than in the aggregated residual stream.

**Compliance With Llm Reviewing Policy:**

Affirmed.

**Final Justification:**

The answers have solved my questions. I would like to change my review to 4. weak accept.

**Key Questions For Authors:**

Soundness:
1. The paper identifies Style Modulation Heads by analyzing cosine similarity and head contribution scores. However, a high contribution score only indicates that the head output is related to the target feature, but it cannot distinguish whether the head generates personality features, the head transmits existing personality features, or the head and other components jointly influence personality features.
2. Steering Coefficient Range: (0.5 ~ 10.0) and (0.5 ~ 14.0)
 2.1 Asymmetrical comparisons may amplify the advantages of Style Modulation Heads.
 2.2 Different layers have different activation scales, so directly comparing the original α values ​​is not reasonable.
3. The paper needs to compare with the performance of random heads in ablation experiments.
4. What is the difference of system queries and evaluation queries(held-out queries)? Why are evaluation queries not shown as system queries?
Significance:
The existing LLM such like ChatGPT and Gemini maintain good textual coherence while preserving the target traits.

**Limitations:**

Yes

**Strengths And Weaknesses:**

This study presents a central topic in LLM controllability research—how to achieve precise behavioral intervention while maintaining generation quality. The paper is clearly written and proposes corresponding solutions to a significant bottleneck in the field of activation intervention (coherence degradation). This paper also offers important insights for future model architecture design and security alignment.

---

> ### Author Rebuttal · Authors · 2026-03-31
>
> We thank the reviewer for the thoughtful and insightful comments.
>
> We revised the figures and added experiments (notably Figures 20) on the following website: https://icml2026-rebuttal-image.vercel.app/
>
> **>W1**
> > it cannot distinguish whether the head generates personality features, the head transmits existing personality features, or the head and other components jointly influence personality features
>
> We agree that a high contribution score alone does not distinguish whether a head generates, transforms, or transmits persona-related features.
>
> Our claim is not that these heads exclusively generate persona. Rather, we show that they are strongly associated with persona formation and serve as effective intervention points.
>
> As shown in **Figure 3**, persona-related signals may already emerge in earlier layers (around L15), while the identified layer (L20) exhibits particularly strong alignment with persona directions. This suggests that these heads may play a role in transforming or amplifying persona-related representations.
> (This claim is common to extended models of Gemma-3-12B and Qwen3-30B-A3B. Please see **W5 of v6it**.)
>
> Our contribution is therefore primarily functional and engineering-oriented: identifying where intervention yields the best trade-off between persona control and coherency.
>
> We also note that the phrasing "persona arise" appears once in the paper, which may have suggested a stronger claim than intended. We will revise this wording for clarity. Importantly, our conclusions do not depend on whether these heads generate or transform persona features.
>
>
> **>W2**
> > directly comparing the original α values is not reasonable
>
> Our evaluation is based on **Pareto trade-offs between trait strength and coherency**, rather than absolute coefficient values. Different intervention locations achieve different maximum levels of trait amplification, and capturing this difference is central to our analysis.
>
> As long as each method is evaluated over a sufficiently wide range to reach its effective operating region, comparison via Pareto curves remains valid and does not depend on the absolute scale of coefficients.
>
>
> **>W3**
> > needs to compare with the performance of random heads in ablation experiments
>
> We thank the reviewer for this valuable suggestion.
>
> We conducted additional experiments where we randomly ablate the same number of heads as the identified style modulation heads. (see **Figure 20**).
>
> We find that random ablation does not significantly affect trait strength, whereas targeted ablation of style modulation heads leads to a clear degradation in trait scores. Regarding coherency, MMLU, and IFEval, both types of ablation (style modulation heads and random heads) retain the performance.
>
> This result provides additional evidence that the identified heads are functionally relevant, rather than being arbitrary high-activation components.
>
>
> **>W4**
> > Why are evaluation queries not shown as system queries?
>
> We believe there is a misunderstanding regarding our setup.
>
> The **system prompt is fixed** across all experiments and defines the persona. Persona vectors are extracted using different user queries under this fixed system prompt.
>
> The **evaluation queries (held-out)** are separate user prompts used only for evaluation, while keeping the same system prompt. This ensures that evaluation measures generalization and is not biased toward the extraction data.
>
> > ChatGPT and Gemini maintain good textual coherence while preserving the target traits.
>
> Regarding deployed frontier models, they rely on prompt-level conditioning, trained by various persona-related prompt conditions and forced to maintain the generation coherency.
> On the other hand, our work studies **activation-level intervention**, which can push the model into out-of-distribution regions and lead to coherency degradation.
> These are fundamentally different settings.

---

> > ### Author Rebuttal · Reviewer_9JDa · 2026-04-01
> >
> > Thank you for the thoughtful response and the additional ablation experiments.
> > W1: The authors clarified that the claim is functional and engineering-oriented, not a strong claim on the "origin" of persona features. This more cautious interpretation is appropriate. The willingness to revise the wording "persona arise" to avoid overclaiming is appreciated. The connection to newer models like Gemma-3 also adds value.
> > W3: The new results in Figure 20 (random head ablation vs. targeted ablation) are crucial. They demonstrate that the identified style modulation heads are not just arbitrary high-activation components but have specific functional relevance to persona expression. This addresses one of my main concerns regarding the specificity of the findings.
> > W4 & Comparison: The authors clarified the misunderstanding regarding system/user prompts. The explanation of why activation-level steering faces different coherency challenges compared to prompt-tuned frontier models (ChatGPT/Gemini) is well-reasoned and highlights the specific difficulty of the "out-of-distribution" problem in activation steering.
> > W2: While comparing absolute coefficients α can be tricky due to different scales of hidden states, the authors' argument that the Pareto trade-off is the primary metric of interest is a fair justification for the experimental design.
> > Conclusion:
> > Overall, the rebuttal resolved the technical ambiguities and provided the requested ablation evidence. While the core methodology remains largely the same, the empirical foundation is now significantly more robust. I am satisfied with the current version and maintain my positive/borderline assessment.

---

> > > ### Author Response · Authors · 2026-04-06
> > >
> > > We sincerely thank the reviewer for the detailed and constructive feedback, as well as for carefully evaluating our rebuttal and additional experiments.
> > >
> > > We are glad that our clarifications and newly added ablations successfully addressed your concerns.
> > > We also appreciate your recognition that the empirical foundation has become significantly more robust.
> > >
> > > Given that the main technical concerns have been resolved and the core contributions are now better supported, we would greatly appreciate it if you could kindly reconsider the current score.
> > >
> > > If there remain any additional concerns or aspects that you would like us to further clarify or strengthen, we would be more than happy to address them.
> > >
> > > Thank you again for your time and thoughtful evaluation.

---

### Official Review · Reviewer_v6it · 2026-03-09

**Soundness:** 3
**Presentation:** 1
**Significance:** 3
**Originality:** 2
**Overall Recommendation:** 3
**Confidence:** 4

**Summary:**

This paper investigates coherency degradation in activation steering for LLMs and proposes a targeted intervention approach. Experiments demonstrate that steering directly on these heads, rather than the full residual stream, yields a substantially better tradeoff between trait expression and text coherency across six personas and two models.

**Compliance With Llm Reviewing Policy:**

Affirmed.

**Key Questions For Authors:**

See weaknesses.

**Limitations:**

yes

**Strengths And Weaknesses:**

**Strengths**

This paper is technically sound, and supported by effective visualizations, comprehensive ablation studies, and a strong evaluation framework.

**Weaknesses**

1. The writing quality is very poor. The main text lacks a "conclusion" section, and the references are extremely informal and poorly organized. They contain a large number of unstructured author names, arXiv links, blog posts, HTML links, and other non-standard sources.
2. The scoring method (Equation 4) does not account for the possibility that heads with large output norms might score highly regardless of their actual persona relevance.
3. The paper does not investigate whether these specific heads also serve other unrelated functions, which complicates the framing of pure "style modulation."
4. The asymmetric coherency finding (positive vs. negative steering degradation) and the performance differences between MLP and Attn Residuals are presented without rigorous theoretical explanations.
5. Experiments are restricted primarily to two similar 7-8B parameter dense models, leaving the approach's effectiveness across different architectures (e.g., MoE) and larger models as an open question.

---

> ### Author Rebuttal · Authors · 2026-03-31
>
> We thank the reviewer for the detailed and constructive feedback.
>
> We revised the figures and added experiments (notably Figures 21 and 22-27), which are available on the following website: https://icml2026-rebuttal-image.vercel.app/
>
> **>W1**
> > The main text lacks a "conclusion" section
>
> While some prior works do not include a dedicated conclusion section [1,2], we acknowledge that adding one can improve clarity and readability. Although the absence of a conclusion does not affect the technical content, we will include a concise conclusion section in the revision.
>
> > They contain a large number of unstructured author names, arXiv links, blog posts, HTML links, and other non-standard sources.
>
> We also agree that the reference formatting should be improved. In the revision, we will revise references to use standardized formats (e.g., proceedings version where available) and remove informal sources.
>
>
> **>W2**
> > (Equation 4) does not account for the possibility that heads with large output norms might score highly regardless of their actual persona relevance
>
> We thank the reviewer for this insightful point.
>
> To address this concern, we computed head importance using only cosine similarity, which removes the effect of output norm. We find that the identified style modulation heads remain unchanged, indicating that large output norms are not driving the results (see **Figure 21**).
>
> We therefore conclude that norm effects do not substantially bias our findings. At the same time, we retain the dot-product formulation in Eq. (4) to account for both direction and magnitude, which better reflects the actual contribution to the residual stream.
>
>
> **>W3**
> > does not investigate whether these specific heads also serve other unrelated functions
>
> Empirically, **Figure 5(b)** shows that interventions at these heads have minimal impact on MMLU and IFEval, suggesting that they are not strongly involved in knowledge retrieval or instruction-following. This supports the interpretation that these heads play a prominent role in persona control, even if they are not functionally exclusive.
>
>
> **>W4**
> > the performance differences between MLP and Attn Residuals are presented without rigorous theoretical explanations.
>
> We agree that a rigorous theoretical explanation is an important open problem.
>
> Our work takes an empirical mechanistic interpretability approach, aiming to identify functional structure through controlled interventions and observations.
>
> That said, we provide an intuitive explanation: MLP layers perform token-wise transformations, while attention layers integrate information across the full sequence. Since persona is a global property of generation, attention mechanisms are more naturally suited for its modulation, whereas MLP residual interventions tend to be less effective.
>
>
> **>W5**
> > leaving the approach's effectiveness across different architectures (e.g., MoE) and larger models as an open question.
>
> In addition to Qwen2.5-7B and Llama-3.1-8B, we conducted experiments on Gemma-3-12B and Qwen3-30B-A3B (MoE) (see **Figures 23-27**).
>
> In both cases, we observe that persona is controlled by a small subset of attention heads, consistent with our main findings.
>
> Interestingly, we find that in larger models, style modulation heads are **distributed across multiple layers**, rather than concentrated in a single layer:
> - Gemma-3-12B: layer 18, 20, 22, 24, 26, 28 (peak effect at L24)
> - Qwen3-30B-A3B: layer 27, 32, 39 (persona-dependent peak)
>
> We also observe that the layer-wise geometric patterns differ from those in Qwen2.5/Llama-3.1. In particular, the transition points in cosine similarity (**Figure 3(a)**) are less clearly localized. In Qwen3-MoE, the persona direction remains largely consistent across layers, while in Gemma-3, transition points are persona-dependent.
>
> We hypothesize that these differences arise from architectural factors such as sliding/global attention and pre-/post-LayerNorm in Gemma.
>
> These results suggest that while the **sparsity of persona-relevant heads is robust, their layer-wise organization depends on model architecture and scale**.
>
> We will include these findings in the appendix and note that a more systematic study is an important direction for future work.
>
>
> [1] Reddy, "The mechanistic basis of data dependence and abrupt learning in an in-context classification task", ICLR 2025 (oral).
>
> [2] Schug et al., "Attention as a Hypernetwork", ICLR 2025 (oral).

---

> > ### Author Rebuttal · Reviewer_v6it · 2026-04-02
> >
> > The author has addressed my concerns.

---

> > > ### Author Response · Authors · 2026-04-06
> > >
> > > Thank you very much for your careful evaluation and for confirming that all concerns have been fully resolved.
> > >
> > > Given that the issues have been adequately addressed, we would greatly appreciate it if you could kindly reconsider the current score in light of this.
> > >
> > > If there are any remaining points that you feel still prevent a higher evaluation, we would be happy to further clarify or improve them.
> > >
> > > We sincerely appreciate your time and careful evaluation.

---

### Official Review · Reviewer_P6aj · 2026-03-09

**Soundness:** 3
**Presentation:** 4
**Significance:** 4
**Originality:** 3
**Overall Recommendation:** 5
**Confidence:** 3

**Summary:**

This paper investigates persona steering in LLMs. Targeted interventions on specific modules and attention heads reveal that a few specific attention heads in specific layers of LLama3 and Qwen2.5 have the most significant impact on persona steering. By intervening on them, output style can be steered most effectively while maintaining performance on other, unrelated tasks.

In an orthogonal analysis, the paper analyses whether using LLM-as-a-judge to score the coherency of generated output correlates with task metrics, such as MMLU or IFEVAL and finds that these task-based metrics are not well predictive of coherency scores. This implies evaluation of steering should rely on LLM-as-a-judge scores more than external task metrics.

**Compliance With Llm Reviewing Policy:**

Affirmed.

**Final Justification:**

My assessment remains that the paper is a timely, well-conducted, interesting, and potentially impactful work. In the rebuttal, additional results have been provided, which I strongly encourage to include in an updated version.

**Key Questions For Authors:**

Overall, my assessment is that this is already a comprehensive and strong submission, with a clear and original contribution towards mechanistically understanding LLMs. Mainly, the claim around insufficiency of task metrics to detect performance degradation caused by steering should be supported with more evidence, as outlines under "Weaknesses".

**Limitations:**

yes

**Strengths And Weaknesses:**

## Strengths
**(S1)** This study is a well-conducted, targeted analysis of the existence of the claimed style modulation heads. Their existence and functional specialization is demonstrated through extensive evidence and ablations. These insights are relevant in understanding the emergence of stylistic traits in model outputs and how to control them.

**(S2)** The paper is overall well written. Explanations and structure are clear and easy to follow, and the visuals are of very high quality. The supplementary material contains extensive additional information on experimental details and results.

**(S3)** The paper is clear about the limitations of the analyzed heads and shows that style modulation heads do in most cases, but not always, yield the best control over stylistic attributes. This is a sound and credible approach towards a topic as complex as mechanistic interpretability of LLMs. It is not expected that we find the "one knob" for a specific behavior.

**(S4)** Experiments on LLM-as-a-judge vs. task metrics could enable better practices in validating steering and are a meaningful contribution to the community.

**(S5)** Heads are identified only through activation geometry, and not through (costly) causal patching or similar techniques. This makes the proposed analysis scalable and practically relevant.

---

## Weaknesses

**(W1)** The claim that conventional task metrics fail to detect coherency degradation of generated text could be strengthened:
  1. In line 216, the paper describes trends of different metrics, but does not (numerically) quantify them through metrics such as correlation or $R^2$ . However, only from the visuals in Fig. 2, these claims are hard to verify, especially, since scale may be an issue.
  2. The paper only evaluates two tasks, MMLU and IFEval. However, other tasks may show higher correlation with LLM-as-a-judge coherency scores. To justify the broad claim that task metrics don't detect quality degradation in outputs (line 089), evaluating a larger and more diverse set of tasks is necessary.
  3. The mismatch of Coherency Score and IFEval score remains unclear. Intuitively, quality degradation in coherency should be strongly correlated with instruction following performance, such as following correct formats. The paper would significantly benefit if this unintuitive finding was explored in more depth.
  4. The relationship between task metrics and coherency is mainly assessed via correlation/predictive performance, but the LLM-as-a-judge scores themselves are validated by human annotators through pairwise comparisons. If LLM-as-a-judge scores can't be directly correlated with scores from human annotators, why can we still draw conclusions from a lack of correlation between task metrics and coherency scores?
  5. The x-axis scales between the first and 2nd row in Fig. 2 are different. This could be confusing to some readers, and overall it is recommended to always show results for the same steering coefficient ranges.

**(W2)** In Fig. 3a), I'm concerned that the geometric patterns (i.e., the three identified blocks), resemble more general phases of LLM processing (e.g. similar to [1]) isntead of style-related patterns. Is there any evidence that the pattern is unique to style/personas?

**(W3)** The paper could discuss more potential applications or use-cases, such as targeted fine-tuning or implementing safety guardrails. Any additions to this would strengthen the paper.

## References
[1] Koishekenov, Yeskendir, Aldo Lipani, and Nicola Cancedda. "Encode, Think, Decode: Scaling test-time reasoning with recursive latent thoughts." _arXiv preprint arXiv:2510.07358_ (2025).

---

> ### Author Rebuttal · Authors · 2026-03-31
>
> Thank you for the detailed feedback and helpful suggestions.
>
> We revised the figures and added experiments (notably Figures 18, 19) on the following website: https://icml2026-rebuttal-image.vercel.app/
>
> **>W1.1**
> > In line 216, ... does not (numerically) quantify them through metrics such as correlation
>
> We agree that quantitative analysis would strengthen the claim.
>
> Beyond the trends shown in **Figure　2**, we observe that commonly used task metrics (e.g., MMLU, PPL) do not consistently track changes in coherency across steering strengths. In particular, while coherency degrades substantially under strong steering, task metrics often remain relatively stable or exhibit inconsistent trends.
>
> We are currently computing quantitative measures (e.g., correlation between task metrics and coherency) and will include these results in the revision to further support our claim.
>
>
> **>W1.2**
> > other tasks may show higher correlation with LLM-as-a-judge coherency scores.
>
> Our goal is to assess whether *commonly used evaluation metrics* in activation steering can reliably capture generation quality degradation. Prior work [1] predominantly relies on MMLU and PPL, and we follow this standard setup. We further include IFEval to broaden the evaluation.
>
> We are currently conducting additional experiments on mathematical reasoning benchmarks and will include them in the revision.
>
>
> **>W1.3**
> > The mismatch of Coherency Score and IFEval score remains unclear.
>
> Our results suggest that instruction-following ability and text coherency are partially decoupled, and degrade at different stages under steering.
>
> Empirically, we observe that as steering strength increases (see **Figure 2**), coherency degrades first, while IFEval remains relatively stable. Under even stronger steering, IFEval also begins to decline, but only after coherency has already collapsed.
>
> This suggests a stage-wise degradation: the model first loses global consistency in generation, and only at higher strengths does it fail to satisfy instruction constraints.
>
> We hypothesize that this is because instruction-following often depends on satisfying local or surface-level constraints, whereas coherency is a global property of the generated sequence and is therefore more sensitive to perturbations.
>
>
> **>W1.4**
> > why can we still draw conclusions from a lack of correlation between task metrics and coherency scores?
>
> We agree that LLM-as-a-judge is not perfectly equivalent to human scoring, and direct alignment with absolute human scores can be challenging.
>
> However, human evaluation on a continuous 0–100 scale is inherently noisy and inconsistent, whereas LLM-as-a-judge provides a more stable and reproducible signal. For this reason, it has been widely adopted as a practical proxy in recent work [2].
>
> Importantly, our claim relies on relative comparisons across steering strengths, rather than absolute score calibration. Even if the proxy is imperfect, the observed mismatch remains meaningful as long as the evaluation signal is internally consistent.
>
>
> **>W1.5**
> > The x-axis scales between the first and 2nd row in Fig. 2 are different.
>
> We thank the reviewer for this suggestion and conducted additional experiments using a unified steering coefficient range. (see **Figure 18**)
>
> We observe that when extending the range to [-4.0, 4.0], differences between metrics become less visually distinct because IFEval begins to degrade simultaneously. In contrast, within a moderate range of [-2.5, 2.5], coherency degradation appears earlier and more clearly.
>
> This observation is consistent with a stage-wise degradation behavior discussed in **W1.3**.
>
>
> **>W2**
> > geometric patterns, resemble more general phases of LLM processing
>
> Persona vectors are constructed as differences between persona-conditioned and neutral prompts, which isolates persona-related features.
>
> To test whether the observed patterns reflect general processing phases, we analyzed hidden states from generic text. (see **Figure 19**)
> We extracted Hidden Vector, where the vector is extracted from the mean response hidden vector of 1000 questions of OpenAssistant without differencing (raw hidden state).
> We find that Qwen shows weaker structure and Llama shows no corresponding transition.
>
> Together with causal results (**Figure 3(b)**), this suggests the pattern is not merely a general processing phase.
>
>
> **>W3**
> > The paper could discuss more potential applications or use-cases
>
> Our findings suggest the existence of domain-specific (e.g., language, medicine, science) attention heads because multi-head attention can be viewed like MoE in terms of dividing one component into multiple roles. Although MLP layers are known to store knowledge, specific attention heads may also capture domain-specific information by sentence-wise processing.
>
>
> [1] Vu et al., "Angular Steering: Behavior Control via Rotation in Activation Space", NeurIPS 2025.
>
> [2] Gu et al., "A Survey on LLM-as-a-Judge".

---

> > ### Author Rebuttal · Reviewer_P6aj · 2026-03-31
> >
> > Thank you for the extensive response to my review. I have read the rebuttal as well as other reviews. As my initial score was already "5 (accept)" I am inclined to maintain this score.
> >
> > ---
> >
> > I have the following further suggestions:
> >
> > > We are currently computing quantitative measures (e.g., correlation between task metrics and coherency) and will include these results in the revision to further support our claim.\
> > > We are currently conducting additional experiments on mathematical reasoning benchmarks and will include them in the revision.
> >
> > I hope the outcome of these experiments confirms the mentioned claims. I appreciate that the authors will state them in their final response.
> >
> > ---
> >
> > > We agree that LLM-as-a-judge is not perfectly equivalent to human scoring, and direct alignment with absolute human scores can be challenging.
> >
> > My main question was rather about the mismatch between LLM Judge vs. humans (pairwise) and LLM Judge scores as a metric (correlation). In my view, it would be better to use the same evaluation mode (pairwise or correlation) for both.
> >
> > Overall it is of course fair to use LLM Judges, but their validity has to be established, too, see [1] for example. I don't see need to provide further experiments on this, but I still would like to bring this aspect to the author's attention.
> >
> > [1] Chehbouni, Khaoula, et al. "Neither Valid nor Reliable? Investigating the Use of LLMs as Judges." The Thirty-Ninth Annual Conference on Neural Information Processing Systems Position Paper Track.
> >
> > ---
> >
> > > W2 of Reviewer v6it (about effect of activation norm)
> >
> > I think this is actually a very important point, but I agree that using cosine similarity and observing the same patterns should clarify it.
> >
> > ---
> >
> > As mentioned above, I will keep my positive rating.

---

> > > ### Author Response · Authors · 2026-04-07
> > >
> > > Thank you very much for your thorough evaluation and for confirming that all concerns have been fully resolved.
> > >
> > > We revised the figures and added experiments again (notably Figures 30-32) on the following website: https://icml2026-rebuttal-image.vercel.app/
> > >
> > > ---
> > >
> > > Regarding the additional experiment mentioned in the rebuttal:
> > >
> > > **>W1.2**
> > > > other tasks may show higher correlation with LLM-as-a-judge coherency scores.
> > >
> > > We conducted additional experiments on GSM8K to examine the relationship between steering strength and task performance(**Figure 30(i), 31(i)**).
> > >
> > > We observe that, unlike coherency, which exhibits asymmetric degradation depending on the steering direction, GSM8K performance degrades more symmetrically.
> > >
> > > Across MMLU, IFEval, and GSM8K, we find that the steering strength at which performance begins to degrade varies across tasks.
> > > As pointed out, it is possible that some tasks may exhibit higher correlation with coherency.
> > >
> > > However, our results suggest that commonly used metrics such as MMLU and PPL alone are insufficient to reliably capture generation quality degradation.
> > >
> > > ---
> > >
> > > **>W1.1**
> > > > In line 216, ... does not (numerically) quantify them through metrics such as correlation
> > >
> > > We additionally computed quantitative correlations between metrics and present a correlation heatmap in **Figure 32**.
> > >
> > > We find that coherency has week correlation with MMLU, but stronger correlation (≈0.7) with PPL and IFEval.
> > >
> > > However, as shown in **Figure 30**, the onset and severity of coherency degradation do not align with changes in these metrics.
> > > It highlights the difficulty of predicting generation quality degradation from task metrics alone.
> > >
> > > We also observe that correlations are generally stronger in Llama than in Qwen, suggesting that Qwen may better disentangle capabilities such as knowledge, instruction-following, and mathematical reasoning.
> > >
> > > We will include these results in the revision to further strengthen the empirical support of our claims.
> > >
> > > ---
> > >
> > > > it would be better to use the same evaluation mode (pairwise or correlation) for both.
> > >
> > > We also appreciate your insightful comment regarding the consistency between LLM-as-a-judge and human evaluation, and will clarify this point in the revision.
> > >
> > > ---
> > >
> > > Thank you again for your time and thoughtful feedback.

---

### Official Review · Reviewer_qAGr · 2026-03-24

**Soundness:** 3
**Presentation:** 3
**Significance:** 2
**Originality:** 2
**Overall Recommendation:** 2
**Confidence:** 5

**Summary:**

This paper aims to navigate the trade-off between controllability and generation coherence in activation steering of LLMs, and conducts a persona steering study of two open-source LLMs that prioritizes coherence over steerability.
Specifically, the authors first hypothesize that the coherence degradation of traditional methods stems from residual stream intervention, which may introduce aggregate noise, and propose to steer the models via their attention heads, which is an approach expected to be more direct and targeted.
The target attention heads are located via geometric analysis of activation spaces using layer-wise cosine similarity and head-wise dot product, and intervention in these heads generally leads to a Pareto front along the coherency-trait axis on two general benchmarks, MMLU and IFEval.

**Compliance With Llm Reviewing Policy:**

Affirmed.

**Final Justification:**

I would like to maintain my concern regarding the fixed intervention mechanism in the paper.

**Key Questions For Authors:**

After reading the main paper, I was still wondering:

1. Why are other component-level interventions less effective?
2. What could be the unrelated noise introduced by residual stream intervention?
3. Which factors determine the upper-bound performance of such activation steering methods?
4. Why did the authors choose coherency instead of coherence? (personally, the latter seems more common?)

**Limitations:**

Yes, in App. F.

**Strengths And Weaknesses:**

### **Strengths**
1. The research problem is clearly stated and a hypothesis is proposed to enhance the soundness of the study.
2. The proposed method achieves a Pareto front along the coherency-trait axis on two general benchmarks and two open-source models.
3. A large volume of analysis results is included in the appendix.

### **Weaknesses**
1. Considering the hypothesis regarding the cause of coherency degradation in LLM steering, more analysis results are expected in the main paper to further demonstrate that the superiority of attention head intervention stems from being more targeted and cleaner than residual stream intervention, rather than from:

    i) the steering vectors being better suited for this location;

    ii) the arithmetic operation (addition) being more suitable for this location; and

    iii) the layer selection being biased toward this location.

2. The method's transferability across layers, personality traits, LLMs, and benchmarks remains largely unclear. Given that the methodology part is relatively simple, it is better to provide more evidence to support its effectiveness and robustness.
3. The authors are encouraged to analyze the effect of attention head number on the steering performance in order to justify their experimental decisions.

---

> ### Author Rebuttal · Authors · 2026-03-31
>
> Thank you for your insightful comments and constructive suggestions.
>
> We revised the figures and added experiments (notably Figures 22-27) on the following website: https://icml2026-rebuttal-image.vercel.app/
>
> **>W1**
> > more analysis results are expected in the main paper
>
> We thank the reviewer for this important comment.
> We clarify that the current paper already controls for these possibilities.
>
> > i) the steering vectors being better suited for this location;
>
> All comparisons use the same steering framework (difference-in-means vector + activation addition), while only the intervention location varies.
> Within the same attention layer, coarse-grained interventions are consistently outperformed by head-level intervention, indicating the gain comes from finer localization rather than vector suitability.
>
> > ii) the arithmetic operation (addition) being more suitable for this location; and
>
> We focus on the standard activation addition setting used in prior work [1]. The same operation is applied across all intervention sites, so the comparison isolates *where* to intervene rather than *how*.
>
> As noted in lines 401-404, while different operators such as Angular Steering [2] may yield different behaviors, this is orthogonal to our contribution, which identifies effective intervention locations under a fixed steering mechanism.
>
> > iii) the layer selection being biased toward this location.
>
> Our goal is to identify the **most effective intervention point**. We therefore focus on layers that are relevant to persona formation (L20 in Qwen, L14 in Llama).
>
> Also, the layer is not selected post hoc. We first identify candidate layers via layer-wise geometric analysis, and then **systematically apply to each layer** (see **Figure 3**) to verify where the effect emerges.
>
>
> **>W2**
> > The method's transferability across layers, personality traits, LLMs, and benchmarks remains largely unclear.
>
> We thank the reviewer for raising this important point regarding generalization.
>
> We would like to clarify that this concern is already addressed in the paper:
>
> - Across layers: As shown in **Figure 3 and Section 4**, we perform systematic layer-wise analysis and intervention, demonstrating that only specific attention layers exhibit strong persona-related effects.
> - Across traits: We evaluate multiple personas including evil, sycophancy, hallucination (from prior work [3]), as well as additional traits (humorous, passionate, loser), all showing consistent behavior.
> - Across benchmarks: Please see **W1.2 in P6aj**.
>
> Beyond this, we further strengthen generality with additional experiments:
>
> - Across models: Please see **W5 in v6it**.
>
>
> **>W3**
> > encouraged to analyze the effect of attention head number on the steering performance
>
> We conducted additional experiments varying the number of style modulation heads (e.g., 1 vs. 2 vs. 3 heads for multiple personas). In all cases, head-level intervention consistently outperforms other locations, indicating that the effect is not dependent on a specific number of heads. (show details in **Figure 22**)
>
>
> **>Q1**
> > Why are other component-level interventions less effective?
>
> As discussed in Section 4.1 (lines 200-213), coarse-grained components (e.g., residual stream or MLP) include many features unrelated to persona.
> This introduces off-target signals that can push the model toward out-of-distribution behavior, leading to coherency degradation.
> In contrast, head-level intervention isolates more relevant features, enabling more stable steering.
>
>
> **>Q2**
> > What could be the unrelated noise introduced by residual stream intervention?
>
> As discussed in lines 210-211, unrelated noise includes factual knowledge [4] and contextual information unrelated to personas.
>
>
> **>Q3**
> > Which factors determine the upper-bound performance of such activation steering methods?
>
> The upper bound of our method corresponds to intervention at the finest functional granularity (attention heads). Further improvements would likely require more fine-grained decomposition such as SAE [5], which needs additional training.
>
>
> **>Q4**
> > Why did the authors choose coherency instead of coherence?
>
> We follow prior work on persona vectors [3], which consistently uses the term "coherency," and adopt the same terminology for consistency.
>
>
>
>
> [1] Arditi et al., "Refusal in Language Models Is Mediated by a Single Direction", NeurIPS 2024.
>
> [2] Vu et al., "Angular Steering: Behavior Control via Rotation in Activation Space", NeurIPS 2025.
>
> [3] Chen et al., "Persona Vectors: Monitoring and Controlling Character Traits in Language Models".
>
> [4] Geva et al., "Transformer Feed-Forward Layers Are Key-Value Memories", EMNLP 2021.
>
> [5] Huben et al., "Sparse Autoencoders Find Highly Interpretable Features in Language Models", ICLR 2024.

---

> > ### Author Rebuttal · Reviewer_qAGr · 2026-04-02
> >
> > My concerns regarding the analysis part remain. Specifically, although this paper demonstrates the attn head intervention is effective in one setting of activation steering, I could hardly agree that this setting is common or standard enough to justify the conclusion about the effectiveness. Thanks for the response.

---

> > > ### Author Response · Authors · 2026-04-07
> > >
> > > Thank you for the clarification and for carefully considering our response.
> > >
> > > We believe the concern mainly stems from how the activation steering design space is structured. We would like to clarify that activation steering methods can be decomposed into three largely orthogonal axes:
> > >
> > > - **How to steer**: the operation used to modify activations (e.g., addition, rotation as in Angular Steering [1])
> > > - **When to steer**: which tokens and with what strength to intervene (e.g., constant vs. dynamic steering [2,3])
> > > - **Where to steer**: which component of the model to intervene on (e.g., residual stream, attention output, or individual heads)
> > >
> > > Our work specifically focuses on isolating the **"where to steer"** axis.
> > >
> > > ---
> > >
> > > Importantly, activation steering is not a niche setting. It is widely studied in mechanistic interpretability [4] and increasingly used in AI safety research [5,6].
> > > In particular, these influential works in this line adopt a consistent baseline setup: activation addition (how), applied with a fixed coefficient across tokens (when), on the residual stream (where).
> > >
> > > Therefore, this setting serves as a de facto standard baseline for analyzing activation steering behavior.
> > > Our work builds directly on this widely used setup, and isolates the *where to steer* axis to investigate whether the commonly used intervention location is indeed optimal.
> > >
> > > As such, understanding its design space is of independent importance.
> > >
> > > ---
> > >
> > > Prior work [1,2,3] has primarily explored improvements along the *how* and *when* axes, while typically fixing the intervention location (most commonly the residual stream).
> > > As a result, the effect of *where to intervene* has remained underexplored.
> > >
> > > Under this perspective, our goal is not to claim that attention-head intervention universally outperforms all possible steering methods, but rather to show that:
> > >
> > > **even within a standard and widely used steering setup (difference-in-means + activation addition with fixed strength), changing only the intervention location leads to a substantially better trade-off.**
> > >
> > > Because the three axes are largely independent, improvements along the *how* or *when* dimensions (e.g., Angular or Dynamic Steering) can be combined with our findings.
> > > However, such methods do not determine *where* to intervene, and thus cannot resolve this aspect of the design space.
> > >
> > > ---
> > >
> > > Therefore, we believe our contribution is complementary and fundamental: identifying effective intervention locations is necessary regardless of the specific steering mechanism used.
> > >
> > > To avoid overgeneralization, we will revise the wording to explicitly scope our claims to this setting.
> > >
> > > We hope this clarification resolves your concern.
> > >
> > > [1] Vu et al., "Angular Steering: Behavior Control via Rotation in Activation Space", NeurIPS 2025.
> > >
> > > [2] Zhao et al., "AdaSteer: Your Aligned LLM is Inherently an Adaptive Jailbreak Defender", EMNLP 2025.
> > >
> > > [3] Wang et al., "Adaptive Activation Steering: A Tuning-Free LLM Truthfulness Improvement Method for Diverse Hallucinations Categories", THE WEB CONFERENCE 2025.
> > >
> > > [4] Zhang et al., "Locate, Steer, and Improve: A Practical Survey of Actionable Mechanistic Interpretability in Large Language Models".
> > >
> > > [5] Arditi et al., "Refusal in Language Models Is Mediated by a Single Direction"
> > >
> > > [6] Limskey et al., "Steering Llama 2 via Contrastive Activation Addition", ACL 2024.

---

### Decision · Program_Chairs · 2026-04-30

**Decision:**

Accept (regular)

**Comment:**

This paper identifies a sparse subset of attention heads (Style Modulation Heads) that govern persona and style in LLMs, and demonstrates that targeted intervention on these heads achieves better controllability-coherency tradeoffs than residual stream steering. The paper also provides an analysis showing that standard task metrics (MMLU, PPL) fail to capture coherency degradation caused by steering.

Reviewer P6aj (5) found the paper timely, well-conducted, and impactful. Reviewer 9JDa updated their score to 4 after the rebuttal, noting that the added ablation experiments (random head vs. targeted ablation) and clarified experimental setup substantially strengthened the empirical foundation. Reviewer v6it acknowledged all concerns as fully resolved but did not update their score of 3. Given the inconsistency between their acknowledgement and score, this review is given a limited weight. Reviewer qAGr (2) maintains the view that the single intervention mechanism (activation addition) is insufficient to generalize the conclusions.

qAGr's concern is noted but is not considered a blocking issue. The authors convincingly argue that "where to steer" is an orthogonal axis to "how to steer," and that activation addition represents the de-facto standard baseline in this literature. Reviewer P6aj, who is also familiar with this area, found this framing reasonable. The concern is better viewed as a direction for future work rather than a fundamental flaw.

Overall, this is a timely and well-executed mechanistic interpretability study with clear practical relevance to AI safety. The AC recommends acceptance.